# Association of a rapidly selected 4.3kb transposon-containing structural variation with a P450-based resistance to pyrethroids in the African malaria vector *Anopheles funestus*

Leon M. J. Mugenzi[1]*, Theofelix A. Tekoh[1,2], Stevia T. Ntadoun[1], Achille D. Chi[1], Mahamat Gadji[1,3], Benjamin D. Menze[1], Magellan Tchouakui[1], Helen Irving[4], Murielle J. Wondji[1,4], Gareth D. Weedall[4,5], Jack Hearn[4,6], Charles S. Wondji[1,4]*

1 LSTM Research Unit, Centre for Research in Infectious Diseases (CRID), Yaoundé, Cameroon,
2 Department of Biochemistry and Molecular Biology, Faculty of Science University of Buea, Buea, Cameroon, 3 Department of Microbiology, The University of Yaounde I, Yaounde, Cameroon, 4 Vector Biology Department, Liverpool School of Tropical Medicine, Pembroke Place, Liverpool, United Kingdom, 5 School of Biological and Environmental Sciences, Liverpool John Moores University, Liverpool, United Kingdom, 6 Centre for Epidemiology and Planetary Health, Department of Veterinary and Animal Science, North Faculty, Scotland's Rural College, An Lòchran, 10 Inverness Campus, Inverness, Scotland, United Kingdom

* leon.mugenzi@crid-cam.net (LMJM); charles.wondji@lstmed.ac.uk (CSW)

**Data Availability Statement:** All genomic data sets are available from the European Nucleotide Archive. Pooled template whole-genome

## Abstract

Deciphering the evolutionary forces controlling insecticide resistance in malaria vectors remains a prerequisite to designing molecular tools to detect and assess resistance impact on control tools. Here, we demonstrate that a 4.3kb transposon-containing structural variation is associated with pyrethroid resistance in central/eastern African populations of the malaria vector *Anopheles funestus*. In this study, we analysed Pooled template sequencing data and direct sequencing to identify an insertion of 4.3kb containing a putative retro-transposon in the intergenic region of two P450s *CYP6P5-CYP6P9b* in mosquitoes of the malaria vector *Anopheles funestus* from Uganda. We then designed a PCR assay to track its spread temporally and regionally and decipher its role in insecticide resistance. The insertion originates in or near Uganda in East Africa, where it is fixed and has spread to high frequencies in the Central African nation of Cameroon but is still at low frequency in West Africa and absent in Southern Africa. A marked and rapid selection was observed with the 4.3kb-SV frequency increasing from 3% in 2014 to 98% in 2021 in Cameroon. A strong association was established between this SV and pyrethroid resistance in field populations and is reducing pyrethroid-only nets' efficacy. Genetic crosses and qRT-PCR revealed that this SV enhances the expression of *CYP6P9a/b* but not *CYP6P5*. Within this structural variant (SV), we identified putative binding sites for transcription factors associated with the regulation of detoxification genes. An inverse correlation was observed between the 4.3kb SV and malaria parasite infection, indicating that mosquitoes lacking the 4.3kb SV were more frequently infected compared to those possessing it. Our findings highlight the underexplored role and rapid spread of SVs in the evolution of insecticide resistance and provide additional tools for molecular surveillance of insecticide resistance.

sequencing data are available under study accessions PRJEB13485 (Malawi 2002 and Malawi 2014), PRJEB24384 (Ghana, Benin, Cameroon and Uganda) and PRJEB35040 (Mozambique 2002, 2016; DRC-Kinshasa and Mikalayi). The sequences of the 5.3kb intergenic region of CYP6P5 and CYP6P9b have been submitted to GenBank accession number: OR000399 and the nucleotide sequence of the 4.3kb SV is provided in supporting file S1 Text. The underlying numerical data for all graphs and summary statistics have been provided as Supporting Information (S6 Table).

**Funding:** This research was funded by the Bill and Melinda Gates Foundation (INV-006003 to CSW) and a Wellcome Trust Senior Research Fellowship grant to CSW (217188/Z/19/Z). The funders had no role in study design, data collection and analysis, decision to publish, or preparation of the manuscript.

**Competing interests:** The authors have declared that no competing interests exist.

## Author summary

*Anopheles funestus* is an important malaria transmitting mosquito whose control relies on pyrethroid insecticides. The ability to control malaria transmitting mosquitoes is threatened by the emergence of insecticide resistance. Developing effective strategies to prevent, slow, or overcome resistance require an understanding of the underlying genetic mechanisms. In this study, we identified a transposon containing insertion in *Anopheles funestus* mosquitoes in Central/East Africa but absent in other regions. Tracking its distribution and frequency over the years revealed a strong selection with frequencies reaching fixation in less than 3 years. Phenotype/genotype association studies revealed a strong correlation between pyrethroid resistance and genetic variant genotype but negative correlation with plasmodium parasite infection in mosquitoes. These findings shed light on the often-overlooked role of structural variations in the evolution of insecticide resistance, offering valuable insights and tools for monitoring and addressing this critical issue in malaria control.

## Introduction

The control of vector-borne diseases, which accounts for more than 17% of all infectious diseases reported in 2020, still greatly relies on using insecticides despite a recently recommended vaccine [1]. Insecticides have effectively reduced the number of cases and deaths due to these diseases, with over 80% of the reduction in malaria cases between 2000 and 2015 attributed to their use [2]. All LLINs currently used (PermaNet 2.0, Olyset, Duranet), including the new generation nets (PermaNet 3.0, Olyset plus and Interceptor G2), contain a pyrethroid insecticide due to their high potency and low toxicity in humans [3]. However, malaria vectors have developed resistance to pyrethroids which has spread widely in field populations [4], severely affecting our ability to control *Anopheles* with evidence that it could be impacting malaria transmission [3,5].

Insecticide toxicity can act as intense selective pressure, leading to the rapid evolution of resistance through the overexpression or modified activity of detoxification enzymes such as cytochrome P450s (Weedall et al., 2019), alteration of the target site [6], thickening of the cuticle [7] and behavioural changes [8]. The widespread insecticide resistance is a major global challenge threatening the efficacy of current and future vector control tools [9]. In the context of pyrethroid resistance, mutations primarily associated with its target, the voltage-gated sodium channel, are common. However, in *An. funestus*, no Kdr mutations have been identified [10]. Therefore, pyrethroid resistance in this species is predominantly metabolic in nature. Most studies on the molecular bases of insecticide resistance have focused on single nucleotide polymorphisms and small indels because they can be readily identified with short reads [11,12]. However, growing evidence shows that structural variants (SVs) contribute to adaptive mechanisms, including insecticide resistance [13].

SVs represent an essential source of genetic variation and are defined as large DNA sequence variations, including duplications, deletions, insertions, inversions, mobile-element transpositions, and translocations throughout the genome [14]. Structural variants are abundant across chromosomes. They are frequently found near genes where they are often associated with expression and likely contribute to phenotypic variations [15] and are predominantly shaped by transposons [16]. Decades of research have shown that the alteration of cis-regulatory regions by SVs can lead to perturbation of gene expression and phenotype [15,17]. In *Anopheles* mosquitoes, gene copy number variations have been identified and correlated with increased expression of insecticide resistance-associated genes [13,18]. A 6.5kb

structural variant in *An. funestus* was recently shown to be associated with the increased expression of two cytochrome P450 genes whose overexpression confers high resistance to pyrethroids [19] in Southern Africa and is absent elsewhere (West, Central and East Africa) suggesting a restriction to gene flow [20]. Population studies have shown a clear demarcation between Southern and Central/Western/Eastern populations with the clustering of Ghanaian, Cameroonian and Ugandan populations [20]. This points to the possibility of resistance-associated mutations not only emerging independently within a population but also spreading from another resistant population as seen for the cytochrome P450s *CYP6P9a* and *b* in southern Africa [21].

A previous study on the promoter region of an insecticide resistance gene *CYP6P9b* in *An. funestus* showed that this gene's 1kb upstream region failed to amplify in certain countries (Uganda and Cameroon) while it was successfully amplified in most regions [22]. Therefore, it was hypothesised that a structural variant in this region could prevent PCR amplification.

In the current study, we investigated whether a structural variant insertion in a cytochrome P450 locus can modify the pyrethroid resistance phenotype in *An. funetus*. We discovered a 4.3 kb insertion in the p450s loci rp1, previously identified by QTL mapping, housing two insecticide resistance genes, CYP6P9a and CYP6P9b [23]. This insertion was found across Central, East, and West Africa but was absent in southern regions. Monitoring in *An. funestus* populations of Cameroon revealed strong selection on this structural variant, suggesting it may be adaptive or linked to nearby adaptive mutations. Genetic crosses indicated a robust association between this variant and resistance to pyrethroids, as well as overexpression of nearby genes.

## Materials and methods

### 1 Mosquito samples

This study involved field-collected mosquitoes and an insecticide-susceptible laboratory strain FANG. Mosquito collections were conducted in four villages in Cameroon, where *An. funestus* s.s. is the predominant vector. The selected villages and corresponding years of collection are as follows: Gounougou (9˚03′00″N, 13˚43′59″E) in 2017, 2020, and 2021; Elende (3˚41′57.27″N, 11˚33′28.46″E) in 2020; Mibellon (6˚46′N, 11˚70′E) in 2018 and 2020; Tibati in 2021 (6˚28' N, 12˚37' E) and Elon 4˚13'49"N, 11˚36'04"E in 2021. Additionally, mosquitoes previously collected in these localities and others were utilized for temporal monitoring and African wide genotyping: Gounougou in 2014 (Menze et al., 2018); Mibellon in 2016 (Menze et al., 2018); Tibati in 2018 2018 [24]; Ghana in 2014 and 2021 [25,26]; Mozambique in 2015; Uganda-Tororo in 2014 [20] and Uganda-Mayuge in 2017; Kenya in 2018; Benin in 2022 and Tanzania in 2018. Indoor resting mosquitoes were collected with Prokopack aspirators between 06:00 am and 9:00 am, following verbal consent from the house owner. The number of blood-fed *Anopheles* females collected varied from one location to another with an average of 100 per day in Gounougou and Elende and 50 or less in some locations like Tibati. The collected blood-fed *Anopheles* females were kept for 5 days until fully gravid before putting them in 1.5ml tubes for forced egg-laying. The $F_1$s were pooled and reared at the Centre for Research in Infectious Diseases (CRID) until the emergence of adults. $F_1$ mosquitoes from Gounougou (collected in 2018) were used for susceptibility testing and evaluating bed net efficacy.

Molecular identification followed the cocktail PCR method of Koekemoer et al. [27] to discriminate members of the *An. funestus* group and confirm the species as *An. funestus* s.s was done. The *An. funestus* s.s laboratory strain FANG, maintained at the insectary of CRID, was used for the crossing and qRT-PCR as the reference strains to determine gene expression.

## 2. Mosquito rearing

Insectaries were maintained under standard conditions at 26 ± 4°C with a relative humidity of 70 ± 10%. Larvae from field and laboratory strains were fed with ground fish food (TetraMin tropical flakes, Tetra, Blacksburg, VA, USA), and adults were provided with 10% sucrose solution on cotton wool.

## 3. Establishing crossing between field and laboratory mosquitoes

The $F_1S$ of field-caught mosquitoes from Elende and Mibellon were crossed with the FANG *An. funestus* colony to allow segregation of the resistance allele already at a very high frequency in the population. $F_1$ larvae from the field mosquitoes and larvae from the FANG colony were reared until the pupal stage, then individually placed in 15ml tubes with 2ml of water and stoppered with cotton wool. This was done to obtain virgin female mosquitoes for each strain. Female $F_1$s of field-collected mosquitoes were crossed with male FANG to generate a field x FANG line using about 500 individuals from each. A reciprocal crossing using about 500 female FANG and 750 $F_1$ males from the field caught mosquitoes was also generated. Only the female FANG crossed with $F_1$ males for the field was successfully established. A restricted cohort of approximately 100 $F_1$ individuals was derived from the initial cross, subsequently subjected to intercrossing to generate the $F_2$ generation. Approximately 200 $F_2$ individuals were obtained and further subjected to crossing and blood feeding to produce the $F_3$ generation, resulting in a population increase to approximately 600 individuals. This F3 population was utilized for the bioassays. The complete study involved the generation of three distinct lines.

## 4. WHO insecticide susceptibility tests

The resistance profile to public health insecticides was determined using the WHO susceptibility bioassays protocol [28]. About 100 female mosquitoes aged 2–5 days old were exposed to the insecticide(s) for either 30 minutes (for the crossing) or 1 hour (for the $F_1$ field population), then transferred to holding tubes and provided with 10% sucrose solution. Fifty mosquitoes exposed to non-impregnated papers were used as a control group. The initial knockdown effect was scored immediately after exposure time, while the mortality rates were scored 24 hours post-exposure. These bioassays were conducted at 26 ± 2°C and 80 ± 10% relative humidity.

## 5. WHO cone assays

The bio-efficacy of several bed nets was evaluated using the WHO cone assay protocol [29]. Cohorts of 50 mosquitoes were exposed to 25cm x 25cm pieces of the netting of PermaNet 3.0 [Combination of 2.8 g/kg of deltamethrin on side panels and deltamethrin (4.0 g/kg) and PBO (25 g/kg) on roof, manufactured by Vestergaard Frandsen], PermaNet 2.0 ($5.5 \times 10{-}5$ kg/m$^2$ of deltamethrin, manufactured by Vestergaard Frandsen), Olyset ($8.6 \times 10{-}4$ kg/m$^2$ of permethrin, manufactured by Sumitomo Chemical), and Olyset Plus ($8.6 \times 10{-}4$ kg/m2 of permethrin and $4.3 \times 10{-}4$ kg/m2 of PBO, manufactured by Sumitomo Chemical). PermaNet 3.0 had 2 pieces from the roof and the side, as the top is impregnated with insecticide and PBO while the side has only the insecticide. Insecticide-free nets were used as controls to which mosquitoes were also exposed. Mosquitoes were exposed to the nets for 3 minutes in 5 replicates (each of 10 mosquitoes) and then transferred into paper cups. The knockdown effect was recorded at 1-hour post-exposure, and the mortality rate was scored 24 hours post-exposure.

## 6. Screening of the structural variant near *CYP6P5/CYP6P9b* from whole-genome sequences from different regions of Africa

SV was searched using PoolSeq data from *An. funestus* populations available under European Nucleotide bioprojects PRJEB24384, PRJEB13485, PRJEB35040, PRJEB35040, PRJEB24351 and PRJEB10294 [18, 20] for 12 populations of *An. funestus* sampled across Africa from 2002 to 2016. The locations and year of collection are Benin (Kpome, 2015); Ghana (Obuasi, 2014); Cameroon (Mibellon, 2015); Uganda (Tororo, 2014); Democratic Republic of Congo [Kinshasa (2015), Mikalayi (2015); Malawi (Chikwawa, 2014); Mozambique [Palmeira (2016), Morrumbene (2002) and Zambia (Kaoma, 2013) plus the FUMOZ and FANG laboratory colonies. These samples underwent sequencing using an Illumina HiSeq 2500 platform ($2 \times 150$ bp, paired-end). These were aligned to the *An. funestus* F3 genome [30] using BWA (v0.7.17-r1188) [31]. Reads were coordinate sorted, and duplicates were marked with Picard (v2.18.15) [32]. The subsequent alignments were visually inspected for evidence of an insertion in the intergenic region of *CYP6P5* and *CYP6P9b* in IGV. In addition, non-reference transposon insertions were searched for computationally using TranSurVeyor (v1.0) [33]. The region with high coverage corresponding to the insertion sequence was submitted to RepBase, assigned the identifier TE-1_AFu, and subsequently annotated utilizing the CENSOR and InterProScan web services [34,35].

### Visualising nucleotide diversity across the rp1 locus

Alignments were re-created with the *An. funestus* F3 genome (VectorBase release 57, Ghurye et al., 2019) for Tororo (Uganda), Mibellon (Cameroon) and Obuasi (Ghana) populations using BWA (v0.7.17-r1188) (Li et al., 2009). Reads were coordinate sorted, and duplicates were marked with picard (v2.27.2) [32]. Pileup files for each population were created using Samtools (v1.6) [31] with minimum phred-scaled base quality of 10 and mapping quality per read of 20. Population genetic statistics nucleotide diversity and Tajimas' D were generated for 1000 bp non-overlapping windows across chromosome 2 using npstat (v1.0) [36] for each population separately. Plots of nucleotide diversity from positions 8,525,000 to 8,575,000 incorporating the *rp1* resistance locus and 4.3 kb insertion site between *CYP6P5* and *CYP6P9b* were created using the karyoploteR package (v1.16.0) in R version 4.0.3 [37].

### PCR amplification and sequencing of the entire *CYP6P5–CYP6P9b* intergenic region

To validate the poolseq results, which indicated the presence of a structural variant in the upstream region of *CYP6P9b*, the entire intergenic region between *CYP6P5* and *CYP6P9b* was amplified. Primers were 6P9dplF CCC CCA CAG GTG GTA ACT ATC TGA A located at 19bp before the stop codon of *CYP6P5* and the 6P9Ra/b TAC ACT GCC GAC ACT ACG AAG located at 35bp after the start codon of *CYP6P9b* and the Phusion high fidelity DNA polymerase (Thermo Scientific) was used for the amplification. The Phusion Taq PCR mix consisted of 3μl of 5x HF buffer, 0.12μl of 25mM dNTPs, 10mM forward and reverse primers, 0.15μl Phusion Taq, 9.71μl of deionised water and 1μl of DNA for 15μl reaction. Thermocycler conditions were: pre-denaturation at 98˚C for 1 minute; 35 cycles of denaturation at 98˚C for 10 seconds, annealing at 62˚C for 30 seconds, extension at 72˚C for 4 minutes; a final extension at 72˚C for 10 minutes. PCR amplicons were stained with Midori Green Advance DNA Stain (Nippon Genetics Europe GmbH) and visualised and size-scored following (1%) agarose gel electrophoresis, using an ENDURO GDS (Labnet) UV transilluminator. PCR products were gel-purified using the QIAquick Gel Extraction Kit (Qiagen, Hilden, Germany), ligated into

PJET1.2 blunt-end vectors and Sanger-sequenced from each end of the 5.3 kb fragment using the plasmid-specific sequencing primers pJET1.2F and pJET1.2R. To sequence the complete fragment, four additional internal sequencing primers were used (S1 Table). Sanger sequencing was done at GENEWIZ; UK. Sequence data were analysed with BioEdit software [38].

## Design of a simple PCR assay to detect the 4.3kb SV and analysis of its distribution and association with pyrethroid resistance

A PCR was designed to discriminate between mosquito samples with the 4.3kb structural variant and those without, consisting of 3 primers. Two primers (4.3kb_INSL_F: GGG GCG CTT TAG TTG AGA T and 4.3kb_INSR_R: CAC GTT TCA AGT GCA GGT GA) form a pair flanking the insertion but, due to the size of the insertion, amplify only for samples lacking the insertion, to produce a 281bp amplicon. A third primer (4.3kb_INS_R: CAT ACG CCT CTC CAG CAT GGA) binding within the structural variant forms a pair with 4.3kb_INSL_F to give a 780bp product only from samples containing the insertion. PCR amplification was done using the Kapa Taq PCR kit (Kapa Biosystems) with a 15μl reaction mix composed of 10x buffer A, 0.75μl of 25mM $MgCl_2$, 0.12μl of 25mM dNTPs, 0.5μM of each primer, 0.12μl of Kapa Taq enzyme, 10.49μl of deionised water and 1ul of genomic DNA. Thermocycler conditions were: pre-denaturation at 95°C for 5 minutes; 35 cycles of denaturation at 94°C for 30 seconds, annealing at 60°C for 30 seconds, extension at 72°C for 1 minute; a final extension at 72°C for 5 minutes. PCR products were visualized as described above. After optimisation, these assays were used to investigate any possible association between this structural variant and pyrethroid resistance using field $F_1$ and laboratory mosquitoes by genotyping mosquitoes dead and alive after insecticide bioassays.

The spatio-temporal distribution of this structural variant in *An. funestus* s.s. populations collected in different location across Africa (Ghana, Cameroon, Kenya, Uganda, Tanzania, and Mozambique) was investigated. Genomic DNA samples from previous collections from across Africa [20] were also used.

## Assessment of the expression of genes in proximity to the 4.3kb structural variant using qRT-PCR

To determine if the 4.3kb SV affected the expression of nearby genes, qPCR was used to compare the expression of 3 genes (*CYP6P5*, *CYP6P9a*, and *CYP6P9b*) found near it. The Elende x Fang cross was used to generate pools of mosquitoes with different genotypes: homozygous for SV (SV+/SV+), heterozygous for SV (SV+/SV-), and homozygous no SV (SV-/SV-). After rearing Elende x Fang mosquitoes to the $F_3$ generation, adult females aged 3–5 days old were collected and kept at -80°C. DNA was extracted from the legs as described previously [19] and used for the 4.3kb insertion genotyping. Bodies were kept in RNAlater (ThermoFisher scientific) and stored at -80°C until genotyping was completed. The bodies were grouped in triplicates of 8 mosquitoes, each according to their genotypes: SV+/SV+, SV+/SV-, and SV-/SV-. RNA was extracted by genotype using the Arcturus PicoPure RNA Isolation Kit (Life Technologies), and cDNA was synthesised using the Superscript III (Invitrogen) as previously described [19]. The qRT-PCR amplification of *CYP6P5*, *CYP6P9a*, and *CYP6P9b* were assessed relative to the susceptible FANG strain on the Agilent MX3005 using the standard protocol from [39]. Primers are listed in **S2 Table**. Relative expression and fold-change of each target genes were calculated according to the ΔΔCT method incorporating PCR efficiency [40] after normalisation with the *An. funestus* housekeeping ribosomal protein S7 (RSP7) and actin 5C genes.

## Identification of putative transcription factor binding sites present in 4.3kb SV

To check the possible role of this SV in gene regulation, we analysed the DNA sequence of the 4.3kb SV using the CiiiDER software [41]. CiiiDER uses the MATCH algorithm to predict the transcription factor binding sites in a query set of DNA sequences. First, the JASPAR CORE non-redundant vertebrate transcription factors [42] were used as the position frequency matrix PFM transcription factor model. Then we selected transcript factors like Aryl hydrocarbon receptor (Ahr), Muscle aponeurosis fibromatosis (Maf), previously implicated in xenobiotic response [43,44].

### *Plasmodium* infection in relation to the 4.3kb structural variant

Genotyping results for Obout 2016 samples revealed the presence of the three genotypes for the 4.3kb SV. Also, previous data had shown that mosquitoes collected in Obout in May 2016 had high *Plasmodium* infection rates. The *Plasmodium* infection rate was 38.7% (72/186), 79.2% of which was *P. falciparum*, 12.5% ovale-vivax-malariae OVM + and 8.3% mix infection [45]. Screening for *Plasmodium* infection using TaqMan assay was done on 186 whole mosquito specimens from Obout. The real-time PCR MX 3005 (Agilent, Santa Clara, CA, USA) system was used for the amplification [46]. Briefly, 2 μL of gDNA for each sample was used as a template in a 3-step program with a pre-denaturation at 95˚C for 10 mins, followed by 40 cycles of 15 sec at 95˚C and 1 min at 60˚C. The primers Plas-F (5´-GCT TAG TTA CGA TTA ATA GGA GTA GCT TG-3´) and Plas R (5´-GAA AAT CTA AGA ATT TCA CCT CTG ACA-3´) were used together with two probes tagged with fluorophores: Falcip+(TCT GAA TAC GAA TGT C) labelled with FAM to detect *Plasmodium falciparum*, and OVM+ (CTG AAT ACA AAT GCC) labelled with HEX to detect *Plasmodium ovale*, *Plasmodium vivax*, and *P. malariae*. *P. falciparum* samples and a mix of *P. ovale*, *P. vivax*, and *P. malariae* were used as positive controls. A sub-set of positive samples was subjected to Nested PCR to confirm and discriminate the species detected by TaqMan based on the protocol of [47]. Only the *P. falciparum* positives and non-infected were used to investigate the *Plasmodium* infection rates in mosquitoes with different genotypes for the 4.3kb SV. In total 79 mosquitoes were analysed.

### Data analysis

The percentage mortality for the bioassays and percentage genotypic and allelic frequencies for the dead and alive groups were computed in Excel 2016 spreadsheet (Microsoft). In addition, the odds ratio statistical test was calculated on the medcalc website (https://www.medcalc.org/calc/odds_ratio.php) to determine the strength of association between genotypes and the ability to survive insecticide exposure. The proportion of each genotype in the alive and dead individuals was determined and then used for the odds ratio calculations. The proportion of the alive individuals were considered as subjects with positive outcome while the dead were considered as subject with negative outcome. Fisher exact test was used to assess the significance level for the differences observed in the bioassay results, and results were considered significant at P value less than 0.05. The null hypothesis (H0) was that there is no statistically significant association between the observed phenotype and the genotype under investigation. Hence any observed differences in the phenotype are due to random chance or factors unrelated to the genotype being studied. Student's t-test was used to compare the means of data obtained for qPCR to determine the level of significance.

## Results

### 1. Identification of 4.3kb structural variant in *Anopheles funestus* populations

Examining the PoolSeq IGV alignments uncovered a significant spike in coverage within the intergenic region in Uganda in 2014 compared to other populations (Fig 1A) corresponding to a 9 bp transposon insertion target duplication at positions 8,556,411–8,556,420 on Chromosome 2, marked by the sequence "CAAATGTACA." While there was faint visual evidence of the insertion in Cameroon, it was absent in other PoolSeq populations. TranSurVeyor confirmed the insertion in Uganda at the same positions but not in Cameroon or any other population. The inserted fragment is found in the reference strain FUMOZ at positions 24,765,013–24,760,696 on Chromosome 3, within the intron of an uncharacterized gene (AFUN019979). Due to the mixed-template PoolSeq approach, accurately determining the frequency of the structural variant (SV) in Uganda and Cameroon in 2014 was not feasible. However, it seems to have been of intermediate to high frequency in Uganda and very low frequency in Camer-

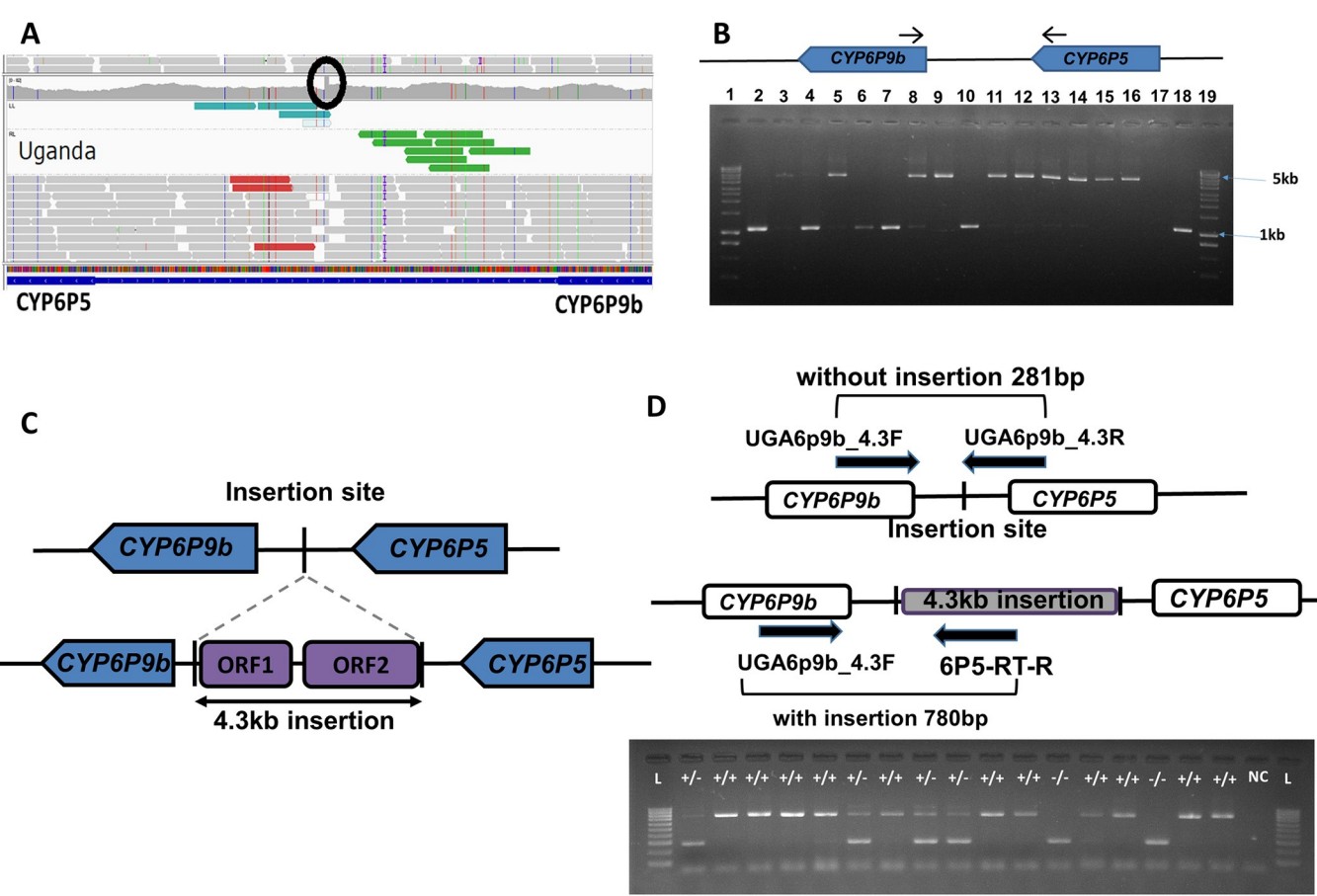

**Fig 1. Identification and genotyping of a 4.3kb structural variant.** A) Screenshot of Uganda individuals from IGV with a bump (circled) where the breakpoints (transposon insertion site) occurred in the Cyp6P5-Cyp6P9b intergenic region. Reads in red represent discordant reads. B) PCR amplification of the CYP6P9b and CYP6P5 intergenic region. Lane1 and 19 are the 1-kilobasepair DNA size marker;2 to 16 are field samples from Cameroon-Mibelong (2016); 17 is the negative control and 18 is a positive control. C) Schematic representation of CYP6P9b and CYP6P5 intergenic region with the structural variant and without the structural variant with the 2 open reading frames identified. D) Schematic representation of CYP6P9b and CYP6P5 intergenic region with the structural variant and without the structural variant Schematic representation of the 4.3kb SV diagnostic assay, consisting of 2 primers flanking the insertion site and 1 in the 4.3kb SV and electropherogram showing the different genotypes. +/+ = SV+/SV+, +/- = SV+/SV- and -/- = SV-/SV-.

oon. In the intergenic region, nucleotide diversity was lower in Uganda compared to Cameroon and FANG datasets, as well as other regions (S1 Fig). This reduction was accompanied by low Tajima's D values (D<0) in the rp1 locus (including the intergenic region between *CYP6P5* and *CYP6P9b* genes) with negative values observed in Uganda, Ghana and Cameroon compared to FANG (S3 Table). These negative Tajima´s D values indicate positive selection or selective sweep across this locus in these populations. Multiple peaks of low diversity in Uganda across the rp1 locus, compared to other populations, suggest complex regional evolutionary patterns not confined to the 4.3 kb insertion. Inverting the ratio of nucleotide diversity between Uganda and other populations revealed a flat profile, indicating little reduced diversity at other rp1 loci versus Uganda (S2 Fig).

## 2. Amplification and sequencing of the 4.3kb structural variant insertion site in Uganda and Cameroon populations

The entire intergenic region between *CYP6P5* and *CYP6P9b* was amplified, cloned and sequenced to confirm the presence of this structural variant in Uganda and Cameroon *An. funestus* s.s. populations. PCR amplicons of approximately 1kb (no insertion) and 5.4kb (with the insertion) (Fig 1B) were cloned into pJET1.2/blunt cloning vectors and Sanger-sequenced. To sequence the full fragment, four additional sequencing primers were used (S1 Table). Sequence reads were aligned to reconstruct the full 5.4 kb region. To improve the sequence, pooled template WGS data were aligned to the draft 5.4 kb intergenic sequence flanked by the two coding genes *(*accession number: OR000399). Results revealed a 4311bp fragment inserted 494bp downstream of the translation stop codon of *CYP6P5* and 494bp upstream of the translation start codon of *CYP6P9b*.

Analysis of the entire 5.4 kb intergenic region revealed the presence of 2 open reading frames corresponding to a putative transposable element. BLASTp identified these open reading frames to match a gag-like protein from *Culex pipens* (AAB86424.1) with percentage identity of 29% and a reverse transcriptase-like protein from *Aedes aegypti* (AAA29354.1) with percentage identity of 39% (Fig 1C). The insertion sequence encodes a retrotransposon in the Jockey superfamily of long interspersed nuclear elements (LINEs). Jockey retrotransposons do not contain long terminal repeats (nonLTR) and are restricted to arthropods in their distribution [48]. The shorter of the two ORFs with 394 amino acids (aas) encodes a zinc finger domain, while the longer domain (906 aas) encodes endonuclease/exonuclease/phosphatase and reverse transcriptase domains in line with other Jockey elements [48]. A CENSOR search in the RepBase database revealed that a Jockey retrotransposon in *An. gambiae* (Ag-Jen-1) as the only similar sequence with a blast alignment score ("bit") of 3,997 and similarity of 0.67. There is a run of 15 Adenine bases in the 3' UTR of the insertion sequence consistent with a polyA tail.

## 3. Design of a genotyping assay for detection of the 4.3kb SV and delimitation of its geographical distribution

The availability of the 4.3kb structural variant sequence and its flanking region facilitated the design of a simple PCR for genotyping its presence or absence in laboratory and field samples. This assay consists of three primers, two flanking this SV and one within it (Fig 1D), as previously designed for a nearby 6.5kb SV between *CYP6P9a* and *b* genes (Mugenzi et al., 2020). The assay was initially tested on Uganda (Tororo, 2014) and Cameroon (Mibellon, 2021) field caught ($F_0$) samples containing the insertion and showed a band at 780bp. Further genotyping of Uganda (Tororo 2014 and Mayuge, 2017) field caught ($F_0$) mosquito samples confirmed its presence and at a high frequency approaching fixation of the 4.3kb SV in Uganda mosquitoes with 100% frequency in Tororo and 97.83% in Mayuge with no mosquito found homozygous for

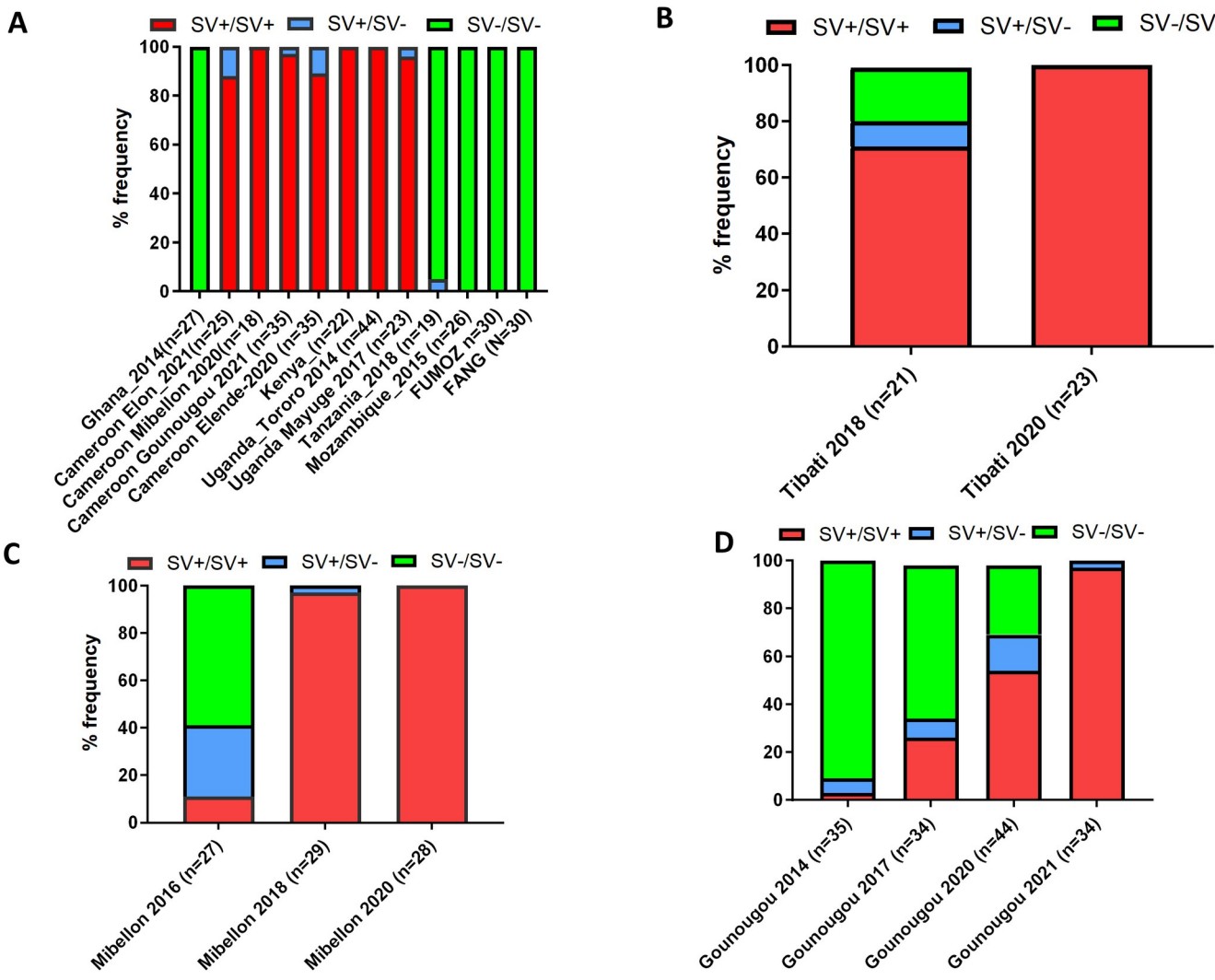

**Fig 2. Spatial and temporal distribution of the 4.3kb SV across Africa.** (A) Geographical distribution of the 4.3kb SV in F0 *An. funestus* samples collected across Africa showing elevated frequencies in Cameroon and Uganda and absence in Ghana and Mozambique. (B), (C) and (D) shows the temporal monitoring of the 4.3kb SV genotypes in Cameroon in (B) Tibati 2018–2020; Mibellon 2016–2020 and (C) Gounougou 2014–2021. All revealing a increase in the frequency of the SV up to fixation.

the wild SV- allele (Fig 2A). Similar results were obtained for the Cameroon mosquito populations collected in Elon (2021), Mibellon (2021), Gounougou (2021) and Elende (2020), with SV+/SV+ genotypic frequencies of 88%, 100%, 97% and 89%, respectively (Fig 2A). Exploring its distribution in other localities across Africa revealed that it was absent from Ghana (West Africa) in 2014 and Mozambique (southern Africa) in 2015 and present at a very low frequency in East African Tanzania (5% SV+/SV-) in 2018 (Fig 2A). Genotyping of the *An. funestus* FUMOZ and FANG lab colonies showed that this structural variant was absent from those colonies.

## 4. Temporal monitoring of the 4.3kb SV allele frequency suggests it has evolved under strong positive selection

The high frequencies of this structural variant observed in Central (Cameroon) and Eastern Africa (Uganda) suggested that it is under a strong positive selection. To investigate this

possibility, we measured the frequencies of this SV in *An. funestus* s.s. populations of Cameroon collected in 2014, 2016, 2017, 2018, 2020 and 2021. Samples were available for 3 locations: Tibati and Mibellon, both from the Adamawa region and Gounougou (Northern region).

In Tibati, a temporal comparison of the frequency of the 4.3kb structural variant in samples collected in 2018 revealed a 76% SV+ allele frequency and complete fixation of SV+ in 2021 (S3A Fig). In addition, all three genotypes (SV+/SV+, SV+/SV- and SV-/SV-) were detected in 2018 at frequencies of 79%, 9% and 19%, respectively, while in 2021, only the SV+/SV+ genotype was detected (Fig 2B), indicating positive selection on the 4.3kb structural variant in this location.

A similar pattern was observed in Mibellon, with the structural variant present at a low allelic frequency (26%) in 2016, increasing to 98% in less than 2 years (2018) and then to 100% in 2020 (S3B Fig). In 2016, mosquitoes without the 4.3kb SV (SV-/SV) were more common (59%) than those with the structural variant in either the homozygous or heterozygous state (39% SV+/SV+; 11% SV+/SV-) (Fig 2C). By 2018 and 2020, the SV-/SV- homozygote was no longer detected, indicating that this 4.3kb structural variant was also driven to fixation under positive selection in this region. Genotyping of this SV in samples collected in Gounougou in 2014, 2017, 2020 and 2021 revealed a similar selection pattern to those observed in Tibati and Mibellon, with 3% in 2014, 31% SV+ in 2017, 63% in 2020 and 98% in 2021. The SV+/SV+ homozygous genotype was at a very low frequency of 3% in 2014, reached a frequency of 26% in 2017 which doubled to 54% by 2020 in about 3 years and then almost got to fixation in 2021 (Fig 2D). The rapid increases in allele frequency of the 4.3kb SV in these 3 regions indicate strong selection acting on this structural variant (or a tightly linked allele) in these mosquito populations.

Analysis of recent samples collected in Ghana and Benin (West Africa) in 2022 showed that this insertion is now present in these localities. In Ghana, this SV was identified at a frequency of 5% with only 3 heterozygotes (S4B and S4C Fig), while in the Benin 2022 samples, a higher frequency of 33% was observed (SV+). The genotypes were present at frequencies of 16% (SV+/SV+), 34% (SV+/SV-) and 50% (SV-/SV-) (S4D and S4E Fig).

## 5. The 4.3kb SV is associated with pyrethroid resistance

We next aimed to establish if this 4.3kb contributes to pyrethroid resistance, thus explaining its swift increase in frequency in field populations of *An. funestus* from Central and East Africa. To investigate this, we employed the following experimental approaches:

1. Genotyping of $F_1$ Gounougou alive and dead field samples from WHO tube bioassays.

2. Genotyping of $F_1$ Gounougou alive and dead field samples exposed to LLINs from cone assays.

3. Genotyping of mosquito progenies issued from field and laboratory crosses exposed to pyrethroids (deltamethrin, permethrin, and alpha-cypermethrin).

Across all experimental approaches, the presence of the 4.3kb SV was consistently associated with increased survival following exposure to various pyrethroids (deltamethrin, permethrin, and alpha-cypermethrin) (Fig 3). The strength of this association was evaluated using odds ratios (ORs) and chi-square tests (Tables 1 and 2). In general, mosquitoes homozygous for the SV (SV+/SV+) exhibited the highest likelihood of survival compared to heterozygotes (SV+/SV-) and those lacking the SV (SV-/SV-). Heterozygotes also demonstrated an enhanced ability to withstand pyrethroid exposure relative to SV-/SV- individuals. The survival probability increased with the number of SV+ copies a mosquito possessed, suggesting an additive

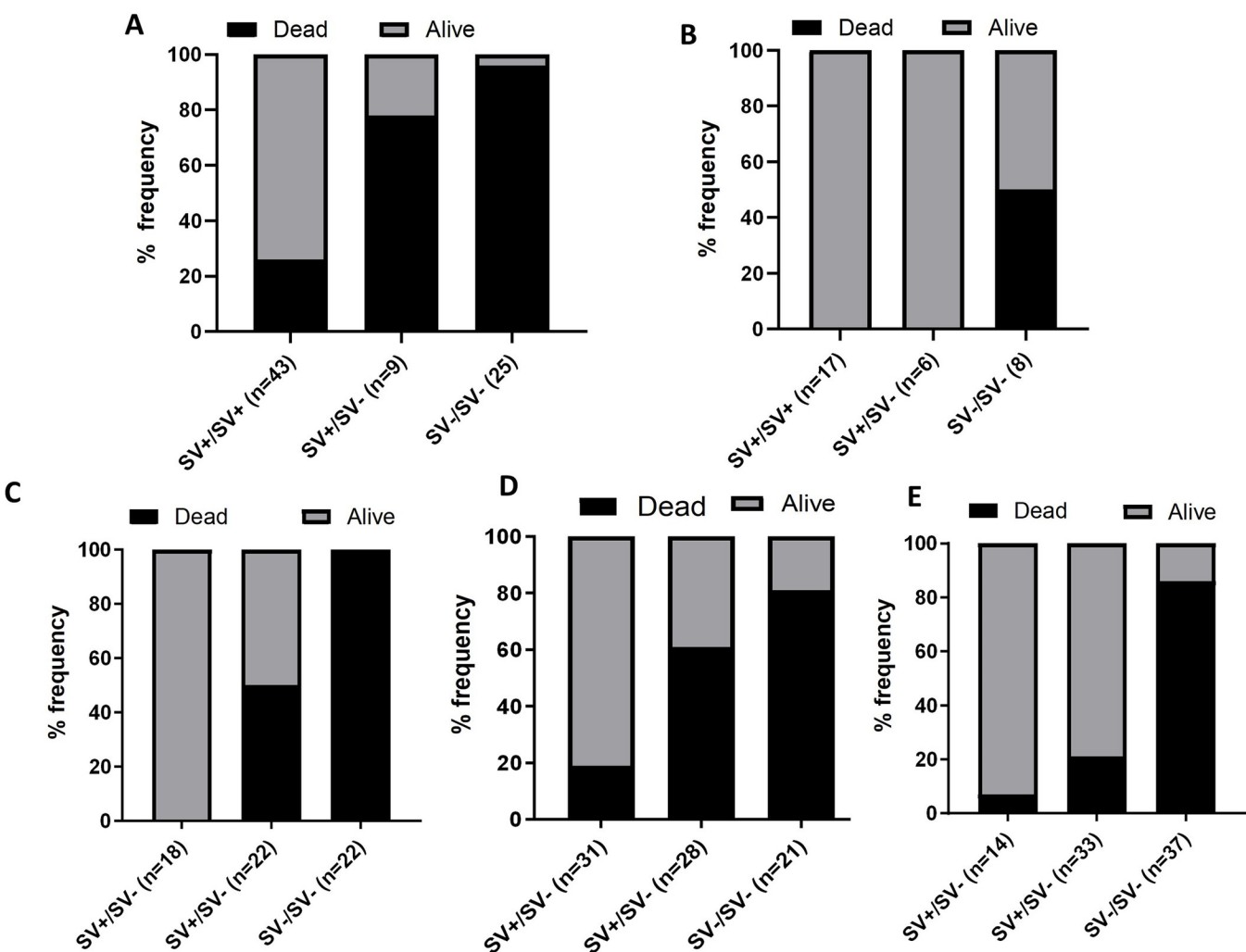

**Fig 3. Association of 4.3kb SV with pyrethroid resistance.** (A) Genotyping results of the 4.3kb SV among the Gounougou 2018 F1s alive and dead after deltamethrin exposure reveal a strong association between the 4.3kb SV and ability to survive. (B) Genotype frequencies of the 4.3kb SV among the Gounougou 2018 F1s alive and dead mosquitoes exposed to PermaNet2 .0 bed nets showing a positive association between the 4.3kb SV and resistance. (C) illustrate the strong association between 4.3kb SV and the ability to survive exposure to deltamethrin by looking at its frequency among dead and alive F3 Elende-Fang crossing mosquitoes. (D) and (E) illustrate the strong association between 4.3kb SV and the ability to survive exposure to insecticide exposure by looking at its genotype distribution among dead and alive F3 Mibellon-Fang crossing mosquitoes exposed to permethrin (D) and α-cypermethrin (E).

effect of the structural variant alleles. At the allelic level, the SV+ allele was significantly associated with survival following pyrethroid exposure, with odds ratios ranging from 1.8 to 116.4 across different experiments and pyrethroids tested. These consistent findings from multiple independent experiments provide strong evidence that the 4.3kb structural variant is associated with resistance to pyrethroids in *An. funestus* populations, likely contributing to the reduced efficacy of pyrethroid-based insecticide-treated bed nets [Detailed results for each experimental approach can be found in the Supplementary Materials S1 Text and S6 Table].

## 6. Impact of 4.3kb structural variant insertion on the expression of nearby genes

To assess potential effect of this structural variant on the expression of nearby genes (*CYP6P5*, *CYP6P9a*, and *CYP6P9b*), crosses between field individuals (Elende, fully homozygous for the

**Table 1. Association between insecticide susceptibility as determined by WHO tube bioassay or WHO cone bioassay and 4.3kb SV genotype in wild-caught, female *Anopheles funestus* from Gounougou Cameroon in 2018.**

| Insecticide/Bednet | Comparison | OR[a] | P value | CI[b] |
|---|---|---|---|---|
| WHO BIOASSAY | | | | |
| Deltamethrin | SV+/SV+ vs SV-/SV- | 69.8 | 0.0001 | 8.4 to 578.5 |
| | SV+/SV- vs SV-/SV- | 6.9 | 0.1380 | 0.5 to 87.3 |
| | SV+/SV+ vs SV+/SV- | 10.2 | 0.008 | 1.8–56.5 |
| | SV+ vs SV- | 8.4 | < 0.0001 | 8.4–41.7 |
| WHO Cone assays | | | | |
| PermaNet 2.0 | SV+/SV+ vs SV-/SV- | 201 | 0.0002 | 12.15–3325.23 |
| | SV+/SV- vs SV-/SV- | 201 | 0.0002 | 12.15–3325.23 |
| | SV+/SV+ vs SV+/SV- | 1 | 1 | 0.02–50.89 |
| | SV+ vs SV- | 82.93 | 0.0021 | 4.98–1379.74 |
| Olyset plus | SV+/SV+ vs SV+/SV- | 4.5238 | 0.3493 | 0.2 to 106.7 |
| | SV+/SV+ vs SV-/SV- | 1.8 | 0.04 | 1.1 to 3.2 |
| | SV+/SV- vs SV-/SV- | 0.3818 | 0.5562 | 0.01 to 9.4 |
| | SV+ vs SV- | 1.8 | 0.04 | 1.0 to 3.3 |

4.3kb SV) and the fully susceptible FANG lab strain (4.3kb SV completely absent) were inter-crossed to the $F_3$ generation. Quantitative real-time PCR performed on pools of each of the 3 genotypes (SV+/SV+, SV+/SV- and SV-/SV-) relative to FANG revealed moderate expression of nearby genes with increased expression of *CYP6P9a* (downstream) and *CYP6P9b* (immediately downstream) but not of *CYP6P5* (upstream) in the SV+/SV+ pool only (Fig 4A). *CYP6P9a* was most expressed in SV+/SV+ with fold change (FC) of 18.7 (*P-value* = 0.008), while SV+/SV- and SV-/SV- genotypes showed no differential expression. Similarly, *CYP6P9b*'s expression was higher in the SV+/SV+ genotype (FC = 16.0, *P value* = 0.002) and low in the SV+/SV- and SV-/SV- genotypes. This indicates that possessing 2 copies of this

**Table 2. Association between insecticide susceptibility as determined by WHO tube bioassay and 4.3kb SV genotype in *Anopheles funestus* crossing between field and FANG lab colony.**

| Insecticide | Comparison | OR[a] | P value | CI[b] |
|---|---|---|---|---|
| Elende X Fang crossing | | | | |
| Deltamethrin | SV+/SV+ vs SV-/SV- | 1517 | 0.0003 | 28.6–80383.1 |
| | SV+/SV- vs SV-/SV- | 41 | 0.01 | 2.2–761.8 |
| | SV+/SV+ vs SV+/SV- | 37 | 0.016 | 2.0–689.9 |
| | SV+ vs SV- | 19.4 | 0.0001 | 9.5–39.7 |
| Mibelong X Fang crossing | | | | |
| Permethrin | SV+/SV+ vs SV-/SV- | 17.71 | 0.0001 | 4.3–72.3 |
| | SV+/SV- vs SV-/SV- | 2.75 | 0.0135 | 0.7–10.4 |
| | SV+/SV+ vs SV+/SV- | 6.44 | 0.0018 | 2.0–20.8 |
| | SV+ vs SV- | 5.63 | 0.0001 | 3.1–10.4 |
| alpha-cypermethrin | SV+/SV+ vs SV-/SV- | 72.72 | < 0.0001 | 18.9–278.8 |
| | SV+/SV- vs SV-/SV- | 24 | < 0.0001 | 10.5–54.2 |
| | SV+/SV+ vs SV+/SV- | 3.05 | 0.009 | 0.8–11.2 |
| | SV+ vs SV- | 116.43 | 0.0001 | 5.5–24.5 |

[a] Odds ratios

[b] confidence interval.

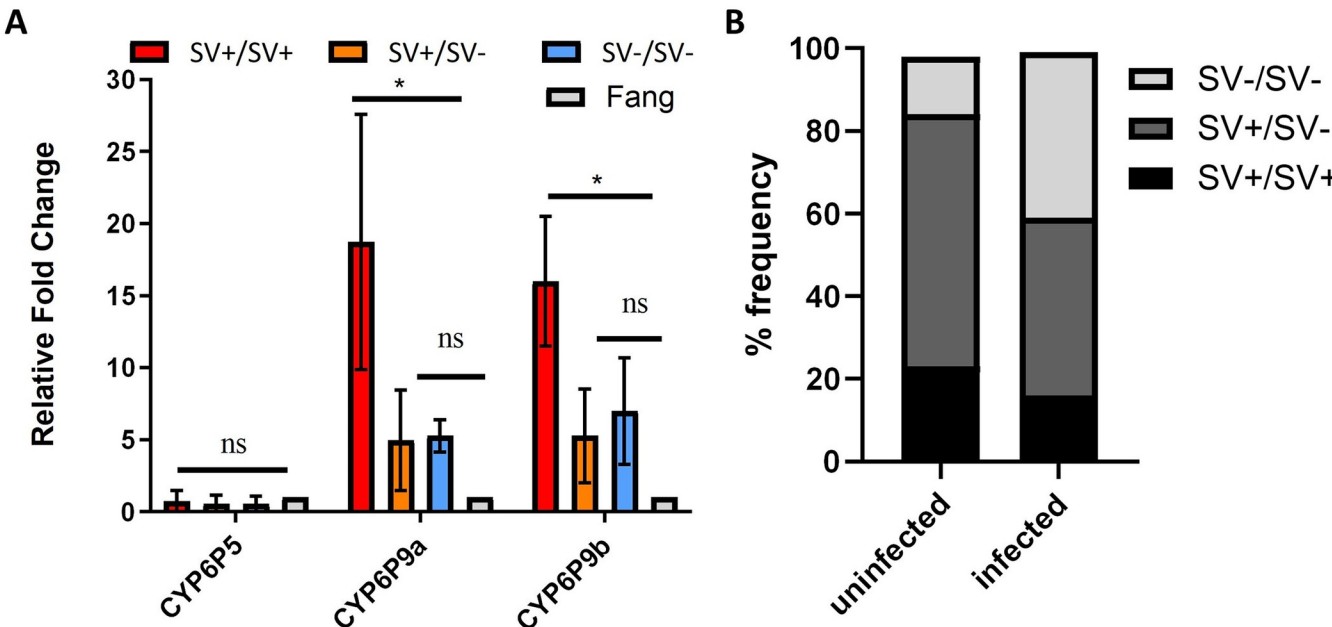

**Fig 4. impact of 4.3kb structural variant on expression of nearby genes and Plasmodium infection.** (A) Differential qRT-PCR expression for different structural variant genotypes for three cytochrome P450 genes in the immediate vicinity of the 4.3kb SV. Error bars represent standard deviation (n = 3). ns = not statistically significant; * = significantly different at p < 0.05. (B) 4.3kb SV genotypes distribution among *plasmodium* infected and non-infected F0 mosquitoes collected from Obout-Cameroon 2016 showing that samples with the 4.3kb SV are less infected than those without the 4.3kb SV.

4.3kb SV enhances the expression of these genes. For *CYP6P5*, there was no difference in the expression level for the different genotypes. Screening for transcription factors known to regulate detoxification genes in the 4.3kb SV using CiiiDER software identified Ahr, ARNT and MAF binding sites. For Ahr/Arnt, 4 binding sites while for 17 MAF binding sites were determined with 3 for MAFG, 2 for MAFF and 9 for MAFb (S4 Table).

## 7. Assessing the association between the 4.3kb structural variant and *Plasmodium* infection

Genomic DNA samples extracted from whole mosquitoes collected from Obout in 2016 were screened to detect *Plasmodium* infection and grouped into 'infected' (n = 37) and 'non-infected' (n = 42). These were genotyped for the 4.3kb structural variant, showing a significant association between genotypes (SV+/SV+, SV+/SV- and SV-/SV-) and infection status ($\chi2$ = 7.0; P = 0.031). The *Plasmodium*-infected group comprised 16.2% SV+/SV+, 43.2% SV+/SV- and 40.5%, SV-/SV-showing that the homozygous SV+/SV+ were less infected than the SV+/SV- and SV-/SV- genotypes (Fig 4B). Hence the SV+/SV+ were less likely to be infected than SV-/SV- (OR: 4.05; CI: 2.2–7.3; P <0.001) while no significant difference was observed between SV+/SV+ and SV+/SV- genotypes (OR: 1; CI: 1.5–474.6; P = 0.05) (Tables 3 and S6). Comparison of the allelic distribution of the 4.3kb SV among infected and non-infected further supported the reduced *Plasmodium* infection in mosquitoes, with 57.8% SV+ being non-infected against 45.2% SV-. Most Plasmodium-infected mosquitoes had SV- alleles without the structural variant (62.1%), while only 37.8% had SV+ alleles (S4 Table). Hence the likelihood of mosquitoes being infected by plasmodium parasite decreases with copies of the SV (Table 3).

**Table 3. Associations between 4.3kb SV genotypes and likelihood of not being infected by Plasmodium parasite.**

| Comparison | OR[a] | P value | CI[b] |
|---|---|---|---|
| SV+/SV+ vs SV-/SV- | 4.1 | 0.0017 | 1.7–9.9 |
| SV+/SV- vs SV-/SV- | 4.2 | 0.0001 | 2.1–8.68 |
| SV+/SV+ vs SV+/SV- | 1 | 0.99 | 0.47–2.1 |
| SV+ vs SV- | 2 | 0.02 | 1.13–3.51 |

[a] Odds ratios, [b] confidence interval.

## Discussion

This study assessed transposon-based structural variations' role in cytochrome P450-mediated metabolic resistance to insecticides in malaria mosquitoes by characterising and studying the effect of a 4.3kb intergenic insertion in a P450 cluster previously associated with resistance in the malaria vector *An. funestus*. A 4.3kb structural variant insertion first detected in East Africa in 2014 was shown to have spread rapidly in Central Africa, notably throughout Cameroon and continuing to move westward on the continent. Analysis of patterns and selection speed of this locus showed its association with strong resistance to pyrethroids, reduced bed net efficacy and *Plasmodium* infection, providing evidence that transposon-based resistance mechanism could be an important driver of metabolic resistance in malaria vectors.

### Rapid selection of transposon-based resistance in malaria vectors with 4.3kb fixation in less than 5 years

PoolSeq data analysis identified this 4.3kb insertion between the *CYP6P5* and *CYP6P9b* loci on Chromosome 2, found only in Uganda and Cameroon samples of 2014 out of 8 countries assessed. This 4.3bk SV contains 2 open reading frames (a gag-like and reverse transcriptase-like proteins), which correspond to 2 (*gag* and *pol*) out of the 3 open reading frames that characterised LTR- retrotransposons. The *gag* encodes capsid proteins, *pol* encodes enzymes regulating the transposition of a mobile element, while the missing *env* encodes a product responsible for the recognition of cell receptors and the penetration of a virus into a cell [49]. The *Drosophila melanogaster* gag-related gene (gagr), a homolog to the Gypsy group of LTR retroelements, is possibly associated with the origin of new functions and the involvement in stress response in Drosophila species [50]. This SV was at a high frequency in Uganda and observed at a low frequency in Cameroon in 2014. Therefore, we hypothesized that this SV spread from Uganda to Cameroon as the likelihood of a retrotransposon inserting twice at precisely the same location appears to be relatively low. Genotyping of recently collected samples for the 4.3kb SV revealed its presence in West Africa in Ghana and Benin at low frequencies, suggesting that this resistance allele is migrating westward. Interestingly population structure analyses using ddRADseq, Poolseq, and microsatellites have revealed a low level of divergence between Cameroon and Ugandan populations of *An. funestus* indicating that there is likely little barrier to gene flow and increased introgression of alleles between them [20,21]. Temporal analysis of the changes in the allelic frequency of the 4.3kb SV in Cameroon collected across the years (2014–2021) revealed a rapid selection of this marker, with its frequency reaching fixation in less than 5 years. Such rapid selection indicates that this insertion or a nearby mutation will likely provide mosquitoes with an essential adaptive advantage. This is supported by the low nucleotide diversity and negative Tajima's D values in the *CYP6P5* to *CYP6P9b* intergenic region (S1 Fig and S2 Table).

The high frequencies observed exhibit a pattern akin to the 6.5 kb structural variant previously identified in *An. funestus* populations from southern Africa, specifically in Malawi and Mozambique [19]. This variant was correlated with an observed increase in resistance to deltamethrin and permethrin in field populations. This 6.5kb SV was shown to increase in frequency from 5% in 2002 to about 90% in 2016 in Mozambique samples [19]. Similarly, in *An. gambiae*, a partial Zanzibar-like transposable element (TE) was identified upstream of *CYP6aa1* in association with two other mutations (nonsynonymous point mutation in *CYP6P4* (I236M) and a duplication of *the CYP6AA1* gene) in Uganda populations at high frequency and shown to have spread to Kenya, the Democratic Republic of Congo and Tanzania [51].

## 4.3kb structural variant is associated with pyrethroid resistance

This structural variant was shown to contain two putative open reading frames. Several transposable elements, including the well-characterized Accord insertion upstream of the *CYP6G1* in *Drosophila melanogaster* that confers resistance to DDT, have been identified in close proximity to metabolic resistance genes [52]. This insertion has spread worldwide in this species [53]. A strong association was found between presence of the 4.3 kb SV and the survival to pyrethroid exposure. Exposing field-collected samples from Gounougou and genetic crosses generated in the insectary, revealed that most of the survivors were homozygous SV+/SV + while heterozygous SV+/SV- and homozygous SV-/SV- were mostly dead. In addition, mosquitoes with the SV+ allele were more likely to survive bed net exposure using cone assays to the standard pyrethroids-only nets (PermaNet 2.0), while PBO-containing bed nets are more effective against these mosquitoes. The results are similar to the association found with the 6.5kb SV even though the association was lower. The odds of surviving exposure to permethrin with the 6.5kb SV was 242.4 (OR) while for the 4.3kb SV it was 5.63 (OR). Further studies are needed in semi-field conditions using experimental huts as previously done for the other metabolic markers [18,19] to better understand the impact of this SV on bed net efficacy.

## A Transposable element containing SV is associated with increased expression of nearby P450s genes and Pyrethroid resistance

Transposable elements are over-represented near P450s clusters in insects [52]. They can affect the expression of the gene(s) by bringing regulatory elements that can either repress or enhance the expression of nearby genes [54]. Genetic crosses were used to compare the expression of three genes located near this 4.3kb SV between the 3 genotypes SV+/SV+, SV+/SV- and SV-/SV- as previously done for the 6.5kb SV [19]. Increased expression of 2 genes (*CYP6P9a* and *CYP6P9b*) was found only in the homozygote SV+/SV+. This higher expression of nearby P450s in the homozygote SV+/SV+ suggests that the transposon contained in this 4.3kb region may act as enhancers. *CYP6P9a* is found at about 4kb downstream of the 4.3kb structural variant. Increased expression was also seen in wildtype RNASeq for Ugandan versus susceptible FANG mosquitoes sampled in 2014 [18] but was less clear cut than the crossing data due to complex expression of *CYP6P9b* across Africa. The RNA sequencing data also unveiled additional P450s exhibiting higher expression levels compared to the ones under our investigation. Therefore, there is a potential that this insertion may be contributing to the increased expression of other genes located at a distance. In *D. melanogaster*, the up-regulation of *CYP6G1* conferring resistance to a variety of insecticide classes [55] correlates with the presence of an Accord retrotransposon in the 5' UTR region and this retro-transposable element contains regulatory sequences capable of increasing the expression of *CYP6G1* in detoxification organs [56]. The 4.3kb SV identified here could regulate the expression of *CYP6P9a/b* by

providing regulatory elements. However, further studies are needed to functionally validate this hypothesis using luciferase promoter assay or promoter assays in transgenic mosquitoes. Additionally, the findings indicate a fully recessive P450 expression phenotype, characterized by overexpression exclusively in SV+/SV+ individuals. The recessive impact of this SV on expression could be due to transvection phenomenon as previously reported in fruit flies [57]. When in heterozygosis (SV+/SV-), a strong repressor in the SV- allele might dilute the activating effect of the 4.3kb SV. For the resistance phenotype, correlation with the presence of the SV+ allele both in the heterozygote and homozygote individuals was observed. Hence the survival of SV+/SV- genotype does not correlate with increased expression of *CYP6P9a/b* genes indicating that other genes might be complementing the observed resistance phenotype. Further investigations, involving bioassays with transgenic mosquitoes overexpressing *CYP6P9a/b* alleles with and without the 4.3 kb SV could help to better understand its contribution to resistance.

### Reduced *Plasmodium* infection in mosquitoes with 4.3kb SV

A negative association was found between the 4.3kb SV and the malaria parasite infection, with mosquitoes lacking the 4.3kb SV being more commonly infected than those with the 4.3kb SV. We might speculate that this is linked to 4.3kb SV-driven overexpression of *CYP6P9a/b* since P450s are known to produce reactive oxygen species [58] which can modulate mosquito immunity to *plasmodium* helping to eliminate the parasite [59]. The role of P450 cytochromes during *Plasmodium* invasion remains poorly understood. Previous studies have shown specific cytochrome P450s implicated in insecticide resistance are also differentially expressed during malaria parasite invasion in mosquitoes [60]. For example, *CYP6M2* and *CYP6Z1* both highly expressed in pyrethroid resistant *An. gambiae* mosquitoes [61,62] were also highly overexpressed in response to *Plasmodium* infection [60], suggesting an interaction between insecticide detoxification and infection. The results obtained are the opposite of what had been previously obtained with the *GSTe2* 119F resistant markers, where an association was observed with the homozygote 119F/F genotype being more associated with the *Plasmodium* infection at the oocyst and sporozoite stage [63]. These preliminary results need further validation using experimental infection studies as previously done with the *kdr* resistance mutations [64] to decipher the impact of P450-based resistance on malaria transmission.

### Conclusion

This study identified a 4.3kb transposon-containing structural variant on chromosome 2 within a cluster of cytochrome P450 genes in Ugandan and Cameroonian populations of *Anopheles funestus* and showed it to be: associated with pyrethroid resistance, associated with enhanced expression of nearby P450 genes; rapidly evolving under strong selection in Cameroon and is spreading westward in Africa. This study shows how genetic variation, such as transposable elements are linked to adaptive changes and rapidly selected in the mosquitoes carrying them. The molecular assay designed here will facilitate the detection and tracking of the spread of this transposon-based resistance and help assess its impact on control intervention and malaria transmission.

### Supporting information

**S1 Fig. Ratios of PoolSeq nucleotide diversity for 1kb windows across the *rp1* locus between two populations closely related to Uganda, Cameroon and Ghana, and the susceptible lab strain FANG.** A peak indicates a loss of diversity in Uganda versus the comparator population. The SV insertion position is labelled with the red-dashed line and gene positions

are indicated below the plots, genes are labelled by their cytochrome P450s name or there VectorBase identifier if not a P450 gene. An upper-bound of 100 was placed on peaks to ensure interpretable plots, these represent windows with very little to zero diversity in Uganda.
(TIFF)

**S2 Fig. Nucleotide diversity (pi) ratios for 1kb windows across the rp1 locus of *An. funestus* between Uganda and Ghana, Cameroon, and FANG PoolSeq datasets.** This is the reverse of S1 Fig in the main text as Ugandan diversity is divided by comparator populations. It shows no peaks indicative of a drop of nucleotide diversity in the denominator population unlike the peaks shown for Uganda in S1 Fig.
(TIFF)

**S3 Fig. (A)** Allelic frequencies of 4.3kb SV in Tibati showing a decrease in SV- allele and increase in SV+ over the time. (B) Allelic frequencies of 4.3kb SV in Mibellon showing a decrease in SV- allele and increase in SV+ over the time.
(TIFF)

**S4 Fig.** Changes in frequencies of the 4.3kb SV over time in (A) Gounougou, (B and C) (D and E) Benin and Ghana.
(TIFF)

**S5 Fig. Association of 4.3kb SV with pyrethroid resistance.** (A) Genotyping results of the 4.3kb SV among the Gounougou 2018 alive and dead deltamethrin post exposure reveal a strong association between the 4.3kb SV and ability to survive. B) illustrate the strong association between 4.3kb SV and the ability to survive exposure to deltamethrin by looking at its genotypic and allelic distribution among dead and alive F3 Elende-Fang crossing mosquitoes. C) illustrate the strong association between 4.3kb SV and the ability to survive exposure to Permethrin by looking at its genotypic and allelic distribution among dead and alive F3 mibellon-Fang crossing mosquitoes. D) illustrate the strong association between 4.3kb SV and the ability to survive exposure to α-cypermethrin by looking at its genotypic and allelic distribution among dead and alive F3 mibellon-Fang crossing mosquitoes.
(TIFF)

**S1 Text. SI results.**
(DOCX)

**S1 Table. Additional sequencing primers were used to sequence the 5.5 kb intergenic sequence.**
(DOCX)

**S2 Table. qRT-PCR Primers.**
(DOCX)

**S3 Table. Tajima's D values in the intergenic region between CYP6P5 and CYP6P9b genes.**
(DOCX)

**S4 Table. Putative transcription factors binding sites in 4.3kb structural variants generate by CiiiDER software.**
(DOCX)

**S5 Table. Allelic frequencies of the 4.3kb SV among plasmodium infected and non-infected mosquitoes.**
(DOCX)

**S6 Table. Genotyping results of 4.3kb in bioassay and Plasmodium infection samples.** (XLSX)

## Acknowledgments

The authors thank the inhabitants of the collection sites for their support during the study.

## Author Contributions

**Conceptualization:** Charles S. Wondji.

**Data curation:** Leon M. J. Mugenzi, Jack Hearn, Charles S. Wondji.

**Formal analysis:** Leon M. J. Mugenzi, Gareth D. Weedall, Jack Hearn, Charles S. Wondji.

**Funding acquisition:** Charles S. Wondji.

**Investigation:** Theofelix A. Tekoh, Stevia T. Ntadoun, Achille D. Chi, Mahamat Gadji, Benjamin D. Menze, Magellan Tchouakui.

**Methodology:** Leon M. J. Mugenzi, Charles S. Wondji.

**Project administration:** Charles S. Wondji.

**Resources:** Helen Irving, Murielle J. Wondji.

**Visualization:** Leon M. J. Mugenzi, Jack Hearn, Charles S. Wondji.

**Writing – original draft:** Leon M. J. Mugenzi, Jack Hearn, Charles S. Wondji.

**Writing – review & editing:** Leon M. J. Mugenzi, Gareth D. Weedall, Jack Hearn, Charles S. Wondji.

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
