## [Decision Letter · Decision Letter 0]

8 Sep 2023

Dear Dr Mugenzi,

Thank you very much for submitting your Research Article entitled 'A rapidly selected 4.3kb transposon-containing structural variation is driving a P450-based resistance to pyrethroids in the African malaria vector Anopheles funestus' to PLOS Genetics.

The manuscript was fully evaluated at the editorial level and by independent peer reviewers. The reviewers appreciated the attention to an important problem, but raised some substantial concerns about the current manuscript. Based on the reviews, we will not be able to accept this version of the manuscript, but we would be willing to review a much-revised version. We cannot, of course, promise publication at that time.

If you decide to revise the manuscript for further consideration at PLOS Genetics, please aim to resubmit within the next 60 days, unless it will take extra time to address the concerns of the reviewers, in which case we would appreciate an expected resubmission date by email to plosgenetics@plos.org.

We are sorry that we cannot be more positive about your manuscript at this stage. Please do not hesitate to contact us if you have any concerns or questions.

Yours sincerely,

Giorgio Sirugo

Academic Editor

PLOS Genetics

Justin Fay

Section Editor

PLOS Genetics

Reviewer's Responses to Questions

**Comments to the Authors:**

Reviewer #1: This manuscript describes the identification of a 4.3kb transposon- containing structural variation and links this to pyrethroid resistance in central/eastern African populations of the malaria vector Anopheles funestus. The results presented are interesting and novel and a cool feature of the paper is the use of temporal collections of mosquitoes to show a rapid increases in allele frequency of the 4.3kb SV in mosquito populations indicative of strong selection acting on this structural variant. Another, very interesting finding was

a negative association between the 4.3kb SV and Plasmodium parasite infection, with mosquitoes lacking the 4.3kb SV being more commonly infected than those with the 4.3kb SV. The manuscript is well written and presented, the methods used are appropriate and the results are well analysed and discussed. I believe this is a valuable contribution to the literature and have only very minor comments for the authors to consider.

In the results it would be useful to have just a little more detail on the transposon implicated in resistance. For example what features) does it contain (in addition to the two ORFs described), are terminal inverted repeats present (if so are these perfect/imperfect)…can the target site duplication site be identified etc.

Line 309: The English in the first section of the results is a little clumsy and should be rephrased to be clearer

Reviewer #2: Mugenzi et al 2023

The paper by Mugenzi et al describes the sequencing of an Anopheles funestus retrotransposon that inserted within a P450 gene cluster, is associated with increased resistance to pyrethroid insecticides, colonizes Africa from East westwards, boosts the expression of one or two CYP genes, contains several binding sites for transcription factors, and decreases adult likelihood to be infected by malaria parasite(s). Or at least, these are the claims.

This work tackles a series of very interesting points, however MS is elliptic on too many key points: retrotransposon identification, qRT-PCR, TF binding sites, which need to be fully clarified. Also, several sections are poorly illustrated, and finally, the way the paper is written is quite too often very far from userfriendly: there is a long list of minor points to correct or improve.

Retrotransposon identification

• Authors confuse ORF, retrotransposon and retrovirus concepts throughout the MS �. The gag and pol ORFs identified in the 4.3 kb SV belong to one retrotransposon, not several. Unlike retroviruses, many retrotransposons do not all contain an env gene (e.g. see the yeast TY1 retrotransposon). Known retrotransposons, in their genomic DNA version, all have either direct terminal repeats (U3/R/U5, e.g. TY1 again), or a 3’ polyA tail (like human LINE elements). Here, such structures have not been searched for.

• Protein homologies should help to clarify the missing retrotransposon classification. Include multiple alignments to gag gene and reverse transcriptase domain of retroelements known to move, from the LTR and polyA classes. This could also be performed for the FUMOZ copy -or copies.

• The described SV may be a full, autonomous retrotransposon or a truncated one, but at any rate, because a target site duplication is shown to exist, the SV inserted through a transposition mechanism, and authors can safely call it a retrotransposon, which would make things clearer. I encourage them to change terminology throughout the MS.

• Submit transposon sequence to a database like repBase

qRT-PCR

• The two CYP6P9a/b genes are highly homologous and several kb apart, casting some doubt on the conclusion that both genes are up-regulated in SV+/SV+ individuals. Specificity controls, for instance with cloned a and b genes should be shown or mentioned. Showing melt curves with different Tm is another option.

• I could not even find a list of used primers, nor PCR conditions…

• Experimental conditions and control data should be included to convince the reader of the observations, and if conclusions are maintained, they should be fully discussed. To my eyes, the negative conclusion on CYP6P5 needs to be tuned down because all expression levels are very low.

• An intriguing outcome of this experiment is that the P450 expression phenotype is fully recessive (overexpression in SV+/SV+ only) while the resistance phenotype is consistently semi-dominant (Fig. 3). This difficulty should be discussed.

•

• The ‘2-ΔΔCT method incorporating PCR efficiency’ (line 261) is self-contradictory. Taking into account PCR efficiencies is a E-ΔΔCT method, where E is the PCR efficiency deduced from a dilution series : E<2 .

• How double normalization is achieved must be explicited¬¬

Putative binding sites for transcription factors (TFs)

• Not only the 4.3 kb SV but the whole intergenic regions (CYP6P5-CYPP9b and CYPP9b-CYP6P9a) should be included in such a hunt.

• Identified candidate target sites should be drawn on a map (possibly Fig1C) and homologies presented in a supplementary Figure.

• The claimed up-regulation of two genes separated by several kb would imply that some TF act in an enhancer, i.e. far away from the TATA box, rather than in a promoter. This is not discussed. Is there evidence for this in the literature for Ahr/ARNT or MAF TFs?

Interference between SV and Plasmodium infection

• A remarkable result, to be mentioned in Abstract.

• Association is weak, although significant, which requires showing odd ratios in a Table

Fig 1A:

• Picture is fuzzy, genomic scale cannot be read etc… Change for a high definition pic imperatively.

• Not everybody is a IVG user… Precise the meaning of colored reads in the caption

• Precise what should be seen in circles.

• Uganda split reads at the retrotransposon insertion site could be highlighted, or at least mentioned.

Fig1C:

• switch also to a high definition picture.

• Precise coordinates of the intergenic region on the reference genome.

• Remove ‘retrotransposonS’, add gag and pol, plus polyA or LTRs if any.

• The position and orientation of the CYP6P9a gene should be drawn, or at least mentioned in Legend.

• Use the same orientation for the top and bottom panels.

• Precise if bottom panel is the reconstituted 5.4 kb, SV containing intergenic region. It does not look like the reference genome, but is just below Fig 1A...

• Place symbols to indicate the relevant TF binding sites identified

Fig 2

• For Fig 2A, precise the pedigree of assayed animals: captured individuals, or their offspring ? It is obscure why the three genotypes are not or cannot be displayed like in Fig 2C, E, F. Avoid “population” in Legend, be more precise

• Give sample numbers.

• Fig2B and D: I could not see usefulness, next to Fig C and E that are perfectly illustrative. Note Fig 2F has no such an appendice, which is fine. Does SV+ refers to the SV allelic frequency, or to the proportion of individuals with at least one copy ?

• Write Cameroun ahead of Tibati, for consistency.

Fig 3

• Fig 3 A,C,E, G,I : stack Alive above Dead consistently. Precise the total number of mosquitoes in each genotype

• Fig 3A and 3C Legend, give insect pedigree: F1 Gounougou 2018 females pooled from several mothers?

• it would be good to adopt the same graphic code as Fig 4B, i.e. 2 collumns with 3 genotypes stacked.

• Remove Fig 3 B,D, F ,H, or explain their added value

Figure 4

• In Fig 4A, the relative expression should be readable as a ratio between genotype of interest and FANG parent. The fold change to FANG must be given for one of the 3 genes and one of the 3 genotypes.

• Remove Fig 4 C or explain its added value: Fig 4B seems sufficient

Table 1

• indicate net active molecule content.

• Include the four net types mentioned in Results

• indicate mosquito numbers and overall mortality %,

• Does SV+ vs SV- refers to a ratio of alleles, or to the proportion of individuals with at least one copy?

• Explicit H0 (presumably no phenotype-genotype association).

Minor points

Abstract

• Abstract should include the identification of TFs within SV, as well as SV role in reducing Plasmodium infection rate.

• L37: Abstract should avoid stat test

• Line 39, expression, not overexpression; tune down the negative conclusion with CYP6P5.

• L39-41, move sentence earlier in the flow of ideas

• The Introduction ends with another version of the Abstract, rather than stating the aims of the study. The Wondj et al 2009 paper does not appear in References.

Introduction

It seems relevant to mention the absence of kdr mutations driving pyrethroid resistance in An. funestus, stressing the importance of metabolic resistance

Mosquito sample rearing:

• L112-114, several publication dates are earlier than the mentioned captures. Explain.

• L118. The offspring of the field caught females is called F1. Indicate the approx. number of females captured, and state whether you pooled the offspring of several females to produce F1 or F3s. I understand that Fig 2A used directly captured individuals rather than their offspring,if so please precise it in M&M.

• L119, precise year.

• L133-142, pedigree is not clear. Do you refer to field caught mosquitoes or to their F1 offspring as parents for the crosses to FANG ? No capture of males nor virgin females from the field is mentioned. If parents for crosses are named F1 individuals, precise the number of insectarium selfings to produce F3s.

• L149 and 160, are there any knock down data in the paper ?

Bioassays

• L155, precise the content of the various insecticide containing nets.

6. SV identification:

• L165-167, redundant with intro, please delete.

• L168-169. Precise sequencing method and the countries/years of origin

Nucleotide diversity

• L189-193, move to legend of Fig S1

Stat tests on genotype segregation

• L298-305. Authors have chosen to explore deviations to expected segregations both by Chi-square tests and by odd ratio stat tests. Please indicate the added value of this double checking. Explicit H0 (presumably no phenotype-genotype association).

Results

• L309-310, precise increase vs what.

• L310-311, replace ‘transposon insertion position’ by ‘ 9 bp transposon insertion target site duplication’,

• L344, replace ‘5.4 kb for the open reading frames’ by ‘5.4 kb intergenic region’.

• L346-348, ORFs, not transposons. Indicate % of homology

• Line 420-424 describes Fig 3A, a 3 genotypes x2 phenotypes matrix, using a 2x 3 matrix. This cannot be easily followed. Either rewrite, or redraw Fig alike Fig 4B.

• Line 424_436: the Result section just parrots data available in Table1, making reading quite reader-unfriendly. The basic conclusion -here, that survival probability increases with SV copy number- is not drawn. This unfortunate style, with conclusions postponed to Discussion, is used repeatedly for all genotype/phe

---

## [Decision Letter · Decision Letter 1]

23 Apr 2024

Dear Dr Mugenzi,

Thank you very much for submitting your Research Article entitled 'A rapidly selected 4.3kb transposon-containing structural variation is driving a P450-based resistance to pyrethroids in the African malaria vector Anopheles funestus' to PLOS Genetics.

The manuscript was fully evaluated at the editorial level and by independent peer reviewers. The reviewers appreciated the attention to an important topic but identified some concerns that we ask you address in a revised manuscript.

We therefore ask you to modify the manuscript according to the review recommendations. Your revisions should address the specific points made by each reviewer.

Yours sincerely,

Giorgio Sirugo

Section Editor

PLOS Genetics

Justin Fay

Section Editor

PLOS Genetics

Reviewer's Responses to Questions

**Comments to the Authors:**

Reviewer #1: The authors have addressed all my comments in the revised manuscript.

Reviewer #2: As said in our initial review, the paper by Mugenzi et al describes the sequencing of an Anopheles funestus retrotransposon that inserted within a P450 gene cluster, is associated with increased resistance to pyrethroid insecticides, colonizes Africa from East westwards, boosts the expression of one or two CYP genes, contains several binding sites for transcription factors, and decreases adult likelihood to be infected by malaria parasite(s).

This work tackles a series of very interesting points, the methodology is quite solid, and the MS has been largely improved. However several points still need improvement.

Inflated claims

The results do not show a causal link between presence of this 4.3 kb transposon and insecticide resistance, but only a strong association. Resistance could be due to a tightly linked mutation, in the rp1 locus for instance. Indeed, Fig S1 shows several rp1 areas with reduced genetic diversity besides the 4.3 kb transposon insertion seen in the resistant Uganda population, suggesting selective sweeps in other areas. These alternative candidate areas for a causal mutation have not been explored, and the 4.3 transposon insertion could be combined to a neighbor causal mutation into an untied haplotype. In the absence of direct evidence, the claim in the title should therefore be tuned down: association, not driving. Idem in the abstract, line 26-27, ‘we demonstrate that the transposon drives resistance…’, and in the conclusion, L723-4.

Another unproven claim is the link between CYP6P9 overexpression and resistance in the studied populations. There are two unsuspected findings:

1) The recessive overexpression of the CYP6P9 genes in the presence of the 4.3 transposon insertion, independently of the resistance phenotype (Fig 4A). This is reminiscent of the transvection phenomenon in Drosophila, which is worth mentioning. For this point, future reporter gene studies in transgenic mosquitoes should be very informative.

2) The semi-dominant resistance phenotype (Fig 3). A semi-dominant or recessive resistance phenotype (L693-6) is not an absolute property, but can vary with the abiotic conditions. Constitutive gene expression data, on the other hand, should only vary with the genotype. The question is not to find conditions in which a semi-dominant phenotype will be hidden, but to explain it when it is found, and results fit only partially, at best, with the ‘more CYP6P9 expression makes more resistance’ explanation (Discussion headline, lines 667-668h). Dissecting the causal chain would require bioassays on transgenic mosquitoes overexpressing CYP6P9a or b in a SV- context, rather than SV-luciferase gene fusions.

To my view, this section of the Discussion (lines 667-696) deserves rewriting.

Data mentioned but not shown

The 4.3 kb SV sequence should be deposited in a public database. It is said to be done, but I could not find it. Please indicate on which line.

L321, the sequence of the FAM and HEX probes should be indicated

L406 SV frequency data for Mayuge not shown

L414-5 SV frequency data for FANG and FUMOZ lines are not shown

L493-4 Data for Olyset and PermaNet3.0 are not shown

Things that must be clarified

The intergenic region length : from Fig1B, the CYP6BP5_CYP6P9b intergenic appears to be ~1kb long in the absence of the 4.3 kb element. However, S2 Table refers to ~50 intervals of 1 kb (probably the whole rp1 locus and not the CYP6BP5_CYP6P9b intergenic region). Please clarify.

L592-3 The described data are not shown on Table 3 and neither congruent with Fig 4B.

L361-362: Tajima’s D values are said to be low, in comparison to what ?? And what should be concluded from this observation ?

Writing

The Result section dealing with Fig3/Table1-2 describes 5 parallel experimental approaches of essentially the same question , in a lengthy description which takes 5 pages and makes the reader seasick with statistics copy-pasted one by one from the Tables. Because the statistical analysis for these 5 variant experiments is the same, and their take home message is basically also the same, describe the 5 variants of the experiment first, and then state the common conclusions from the statistic tests once and only.

The various pyrethroids/nets used are not appearing in the Discussion. Including their composition in Table 1 or in a Supp Table would help.

L69 explicit LLINs

L216 Visualizing diversity across the rp1 locus ?

L267 repeat of line241 : to delete

L295, The phrasing of ‘the 2-��CT method incorporating PCR efficiency’ is self-contradictory (“The ��CT method incorporating PCR efficiency” would be fine).

L334, check syntax

L387, 390-1 check syntax

L462-3 fuzzy writing: association to WHAT of the SV ?

L480-1 increase is not significant from 0 to 1 copy

L547-8 Syntax ??

L632 : this insertion or a nearby mutation

L647 comes back on an hypothesis judged very unlikely on line 622: really no need for this…

L676-7, the 4,3 kb region contains a single transposon only. Since it might contain an enhancer, the distance between CYP6P9a and the 4.3kb insertion should be mentioned.

Line701-703, P450s are not mentioned in the NO production nor oxidative stress in the two quoted papers. It could be recalled that the reduction in viral load associated with SV+ allele, although significant, is only modest.

Figures:

On Fig1C, the 4.3 kb insertion appears to overlap partially the CYP6P9b gene.

Table S3/ Fig S1: if genomic stats were computed on the same intervals (as said L 222-224), why show the output separately rather than stacked on a single Figure ?

Also, it may help to read the raw numbers of dead/alive mosquitoes on the Fig3 and Fig4B histograms

S3Fig is not called in Results

Reviewer #3: All my comments have been adequately addressed

**Have all data underlying the figures and results presented in the manuscript been provided?**

Reviewer #1: None

Reviewer #2: **No: **Could not find the sequence of the 4.3 kb transposon described

Reviewer #3: Yes

PLOS authors have the option to publish the peer review history of their article (what does this mean?). If published, this will include your full peer review and any attached files.

Reviewer #1: No

Reviewer #2: **Yes: **Jean-Marc Bonneville

Reviewer #3: No

---

## [Editor Report · Decision Letter 2]

17 Jun 2024

Dear Dr Mugenzi,

We are pleased to inform you that your manuscript entitled "Association of a rapidly selected 4.3kb transposon-containing structural variation with a P450-based resistance to pyrethroids in the African malaria vector Anopheles funestus." has been editorially accepted for publication in PLOS Genetics. Congratulations!

Yours sincerely,

Giorgio Sirugo

Section Editor

PLOS Genetics

Justin Fay

Section Editor

PLOS Genetics

Comments from the reviewers (if applicable):

**Data Deposition**

http://datadryad.org/submit?journalID=pgenetics&manu=PGENETICS-D-23-00856R2

**Press Queries**

---

## [Editor Report · Acceptance letter]

24 Jul 2024

PGENETICS-D-23-00856R2 

Association of a rapidly selected 4.3kb transposon-containing structural variation with a P450-based resistance to pyrethroids in the African malaria vector Anopheles funestus. 

Dear Dr Mugenzi, 

We are pleased to inform you that your manuscript entitled "Association of a rapidly selected 4.3kb transposon-containing structural variation with a P450-based resistance to pyrethroids in the African malaria vector Anopheles funestus." has been formally accepted for publication in PLOS Genetics! Your manuscript is now with our production department and you will be notified of the publication date in due course.

With kind regards,

Jazmin Toth

PLOS Genetics

On behalf of:
